# Local adaptation to climate anomalies relates to species phylogeny

Yolanda Melero [1,2✉], Luke C. Evans[1], Mikko Kuussaari[3], Reto Schmucki [4], Constantí Stefanescu[5], David B. Roy [4] & Tom H. Oliver [1]

Climatic anomalies are increasing in intensity and frequency due to rapid rates of global change, leading to increased extinction risk for many species. The impacts of anomalies are likely to vary between species due to different degrees of sensitivity and extents of local adaptation. Here, we used long-term butterfly monitoring data of 143 species across six European bioclimatic regions to show how species' population dynamics have responded to local or globally-calculated climatic anomalies, and how species attributes mediate these responses. Contrary to expectations, degree of apparent local adaptation, estimated from the relative population sensitivity to local versus global anomalies, showed no associations with species mobility or reproductive rate but did contain a strong phylogenetic signal. The existence of phylogenetically-patterned local adaptation to climate has important implications for forecasting species responses to current and future climatic conditions and for developing appropriate conservation practices.

[1] School of Biological Sciences, University of Reading, Whiteknights, PO Box 217, Reading, Berkshire RG6 6AH, UK. [2] CREAF, E08193, Bellaterra, Cerdanyola del Vallès, Catalonia, Spain. [3] Finnish Environment Institute (SYKE), Biodiversity Centre, Latokartanonkaari 11, FI-00790 Helsinki, Finland. [4] UK Centre for Ecology & Hydrology, Wallingford, Oxfordshire OX10 8BB, UK. [5] Museu de Ciències Naturals de Granollers, Francesc Macià 51, Granollers, Catalonia 08402, Spain. ✉email: yolanda.melero@reading.ac.uk

Climate change threatens many species with population decline and higher extinction risk, not only through the effects of increases in mean temperature[1–3] but also due to the increased intensity and frequency of climatic anomalies (i.e., variations from average climatic values[4,5]). For several species, climatic anomalies may have larger impacts than continuous increases in global mean temperature[5,6].

Bioclimatic studies have reported population responses to micro-to-macro scaled climatic events[7,8], but none have tested for contrasting population dynamics across the species distributional range due to adaptations to the local biotic and abiotic conditions. Here, we hypothesize a continuum describing the degree of local adaptation for different species, from those showing relatively higher sensitivity to anomalies in the local-versus-global climatic regime ("locally adapted" species) to those with higher sensitivity to global anomalies ("globally adapted"). This corresponds to the known trade-offs between performance and tolerance, with populations of locally adapted species performing best at the average local climatic conditions, but being sensitive to the local anomalies, hence showing population declines below and above the optima; while globally adapted species will best perform around the average conditions occurring across their distributional range, and being sensitive more to the anomalies around this average (Fig. 1). Understanding the degree of local adaptation is essential for predicting species responses to climate change[9,10]. To date, most research has focused on global or continental species responses to climate change[11,12], while local responses have received less attention[13,14]. Comprehension of species' spatial adaptations at the appropriate scale is needed and likely to lead to a better understanding of current species trends and improve predictions from bioclimatic models.

Previous research on the phenological responses of butterflies provides some evidence of local adaptation across species[13], with similar findings from single-species studies[15]. It is likely that species attributes contribute strongly to adaptive potential[9,13] and subsequent responses to global change[16,17]. Therefore, we expect that life-history traits mediate the degree of local adaptation to climatic anomalies. Experimentally designed research on insects has shown that local adaptation to climate is mediated by both physiological[18,19] and morphological attributes (e.g., fur on body and melanin pigmentation[20]). However, it is unclear how these results translate to species population dynamics at broad spatial scales; partly due to the constraints of gathering long-term data across the large distributional ranges of multiple species[14].

In this study, we performed a comparative analysis using multiple species of butterflies as a study system because, like many other thermophilous organisms, they are highly sensitive to weather and strongly affected by climate change[21–23], and show rapid responses to changes[24], minimizing (in comparison to plants and birds, for example) the demographic time lag to extinction[25,26]. Moreover, they show broad variation in life-history traits[27], and have been subject to spatially- and temporally replicated standardized abundance sampling across a large continental extent.

We first evaluate if species responses can be classified on our hypothesized continuum of local-global adaptation. Based on a simulation that demonstrates our hypothesis (Fig. 1), we expect that species adapted to local climatic conditions have population dynamics best explained by local rather than global climatic anomalies, i.e., they are more sensitive to anomalies occurring at the local spatial scale in terms of higher goodness of fit between interannual population change versus anomaly from average local climate conditions. Second, we test if local or global species' responses to climatic anomalies are mediated through key butterfly species attributes.

We hypothesized that the relative sensitivity to local climate anomalies (our proxy of local adaptation) would be higher for species with less gene flow between populations and with higher reproductive rates. Therefore, we expected increased adaptation to local weather anomalies for (i) less mobile species, and (ii) species with higher reproductive rates (assessed in terms of the number of generations per year). We found a wide range of responses across species, with some more sensitive to climatic anomalies at the local site level, to others more sensitive to climatic anomalies calculated relative to a global mean across the whole dataset. We also found this interspecific variation was not related to species mobility or reproductive rate, although it was phylogenetically patterned.

## Results and discussion

**Population change and climatic anomalies**. To test our hypotheses, we used count data from 143 species collected across 172 sites within six European bioclimatic regions between 1999 and 2017 (167,288 site-year-species data points). For each site, butterfly counts were extracted from one of three European butterfly-monitoring schemes, all using a standardized transect-based methodology (the "Pollard Walk"[28]). We used the weekly counts collected across six bioclimatic regions to derive species' regional phenologies (flight period) and calculate series of annual abundance indices for each species and each site[29]. Specifically, while accounting for density dependence (known to be important in butterfly populations[30–32]) we modeled how annual population change per species varied in response to climatic anomalies at two spatial scales: local (i.e., climatic anomalies in relation to the average across all years at that specific site) and global (i.e., climatic anomalies in relation to the average climate across all years and all sites where the species is found).

Climatic anomalies were calculated for temperature, precipitation, and aridity occurring during species-specific phenological periods (overwinter, pre-flight, flight, or post-flight period), both for the year of population change $t$ (except post-flight period) and the previous year $t – 1$, thereby accounting for possible direct and delayed population responses. Through AIC-based model selection (Supplementary Data 1), we identified 86 species for which population dynamics were significantly explained by climatic anomalies, of which half were affected by anomalies in temperature ($n = 48$), either at the local or to the global scale, and 38 species responded to anomalies in precipitation or aridity. This result unifies findings from previous studies which have related changes in populations of butterfly species individually to either temperature[25], precipitation[21], or aridity[33,34]. Overall, the importance of climatic anomalies at the most sensitive phenological periods varied between the species, indicating idiosyncratic interspecific variation in weather sensitivity[2,5] (Supplementary Data 2).

**Interspecific variation and the mediating role of species attributes**. To test our first hypothesis, we quantified the extent to which species were either locally or globally adapted by comparing the variance in year-to-year population change explained ($R^2$) for each species using either local or global climatic anomalies, using the most appropriate climate variable for that species (see above). The degree of apparent local adaptation for each species was calculated as the difference between the two $R^2$ values, ranging from −1 to 1, with higher values indicating greater relative sensitivity to local climatic conditions, and suggestive of local adaptation (Fig. 1). Despite some species showing a low degree of apparent local adaptation (<0.05), we observed a continuum of adaptation between species, from most locally adapted (e.g., *Brenthis ino*, *Melitaea parthenoides*) to most globally adapted (e.g., *Cupido osiris*, *Apatura ilia*; Fig. 2; Supplementary Data 1 and Data 2). Species at the ends of the continuum

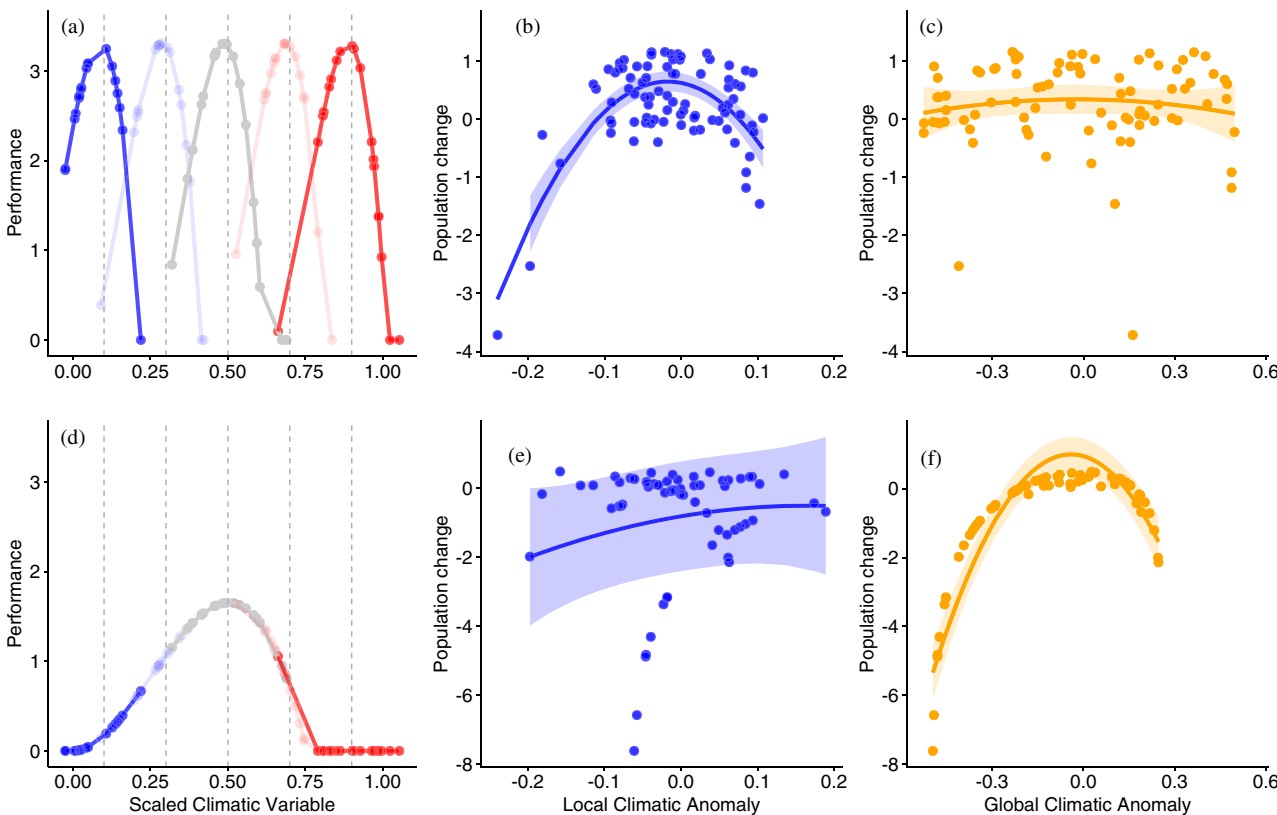

**Fig. 1 Simulations of the consequences of global and local adaptation on the population responses to local and global climatic anomalies. a**, **d** show the performance of species at five sites with climatic means spanning across the range of the climatic variable. We expected locally adapted species to present multiple different performance curves representing distinct populations at sites distributed along the species' distributional range, as shown in panel **a**. This expectation implies that population change will be more sensitive to local weather anomalies (simulation in **b**) than to weather anomalies calculated from all sites across the species' distribution (simulation in **c**). In the case of global adaptation, performance is represented by a single curve through its entire range (**d**). Therefore, observed population change will be more sensitive to global weather anomalies calculated from all sites across the species' distribution (**f**) than to the local site anomalies (**e**). Performance curves were based on a Briere type I function[74], which is a simple function that matches empirical data on thermal performance[75]. We included a fixed area under the curve as consistent with expectations of specialist/generalist trade-offs[76]. Beyond this hypothetical example, in practice, the mean and variance of curves may vary across species; for example, some species may have broader climatic tolerances than others, i.e., the curves in panels **b** or **f** would be broader and shallower, meaning there is a greater range of temperatures in which population growth can remain positive. Broader tolerances may be driven in part by phenotypic plasticity, i.e., gene-by-environment interactions (for example, oviposition microsite preferences varying between locations depending on the local macroclimate). This phenotypic plasticity may be exhibited across the entire range, or it might only occur in certain areas, i.e., there might conceivably be a local adaptation of the phenotypic plasticity. Alternatively, local adaptation can occur in fixed traits, such as lighter-colored insects in warmer areas[77]. Both of these evolutionary adaptations produce patterns akin to that in panel a, whereby optimum of a thermal performance differs amongst the populations of species so that they perform best in their "home" conditions. To generate weather across the range, we standardized an observed 19-year time series of global yearly temperatures (min = 0, max = 1, mean = 0.5) and then shifted the values of each year to predict mean expectation at local sites across the range, a local value for each site and year was then sampled with Gaussian noise. The performance was subsequently used as input into a discrete logistic growth model ($N_{t+1} = RN_t(1 - N_t/K)$) as proportional to R the intrinsic growth rate. Each population was seeded with a small number of individuals and was allowed to recover by immigration should the population size go to zero. A time series of population change for each of the sites was collected from the simulation (ΔN after initialization and immigration was excluded). Models for population change were then fitted using local and global anomalies and are shown in (**b**, **c**, **e**, **f**). Colors in (**a**, **d**) indicate location in the distributional (e.g., blue to red, cold to hot extremes, respectively). Colors in the rest of the panels indicate spatial scale (blue—local climate anomalies; orange—global climate anomalies), circles indicate populations (i.e., from distinct sites), lines show predicted trend with 95% confident intervals.

responded much more strongly to anomalies at a specific scale (local or global), producing unimodal "n-shaped" responses, which are expected given sampling across their climatic niche (illustrative examples provided for *Brenthis ino* and *Cupido osiris* Fig. 3; Supplementary Figs. 1–86).

For the second hypothesis, we tested if relative sensitivity to local-versus-global climate anomalies was mediated by species' mobility[35] and reproductive rate (number of generations per year; i.e., voltinism, with species categorized as univoltine versus multivoltine[36]). We also accounted for the potential effect of

their phylogenetic relationships (see "Methods"). We found a strong phylogenetic signal, explaining 84% of the variation (Pagel's lambda = 0.84; *P* value < 0.0001; Fig. 4), suggesting phylogenetic patterning in the degree of local adaptation across species (Fig. 1). However, we found no effect of either mobility or reproductive rate (Supplementary Fig. 87 and Supplementary Table 1); although mobility and reproductive rate had a significant phylogenetic signal, this did not explain the degree of species local adaptation beyond that expected from taxonomic relatedness (Supplementary Fig. 88 and

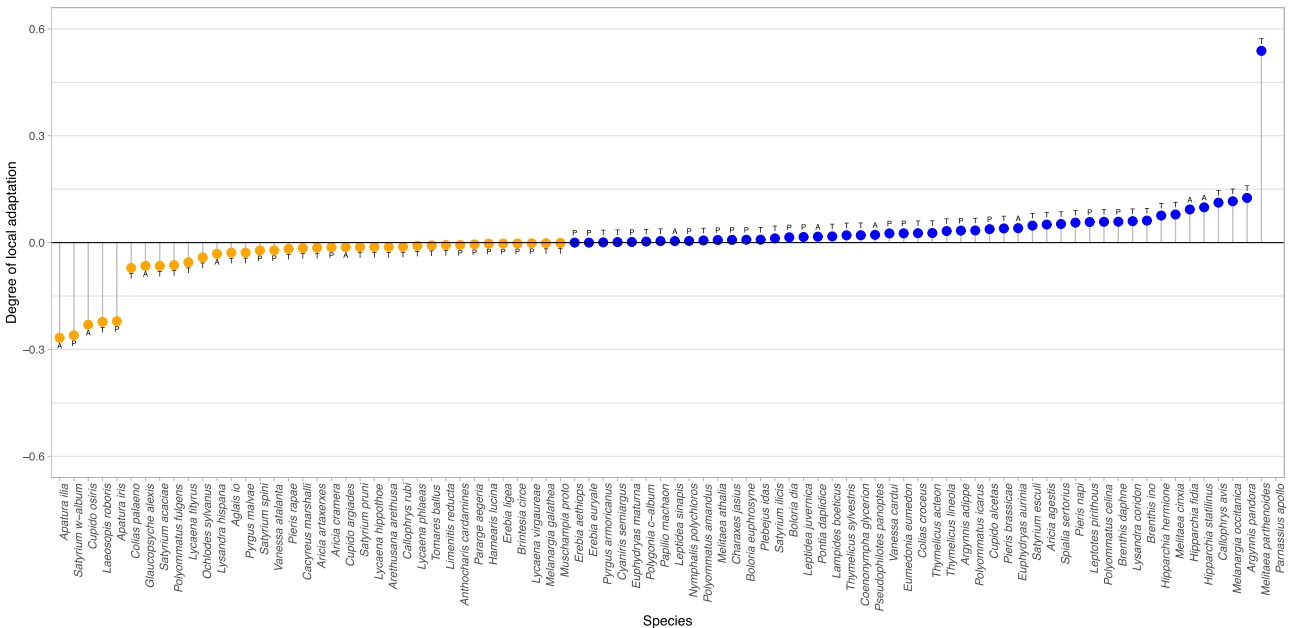

**Fig. 2 Degree of apparent local adaptation across European butterfly species, determined by relative sensitivity of population dynamics to local-versus-global climatic anomalies ($n = 86$).** Degree of local adaptation is calculated as the difference between the $R^2$ of the model including the climatic anomaly at the local scale (site-specific) and the model including that correspondent climatic anomaly at the global scale (all sites), with positive values (blue points) indicating greater local adaption and orange points indicating greater global adaptation. Letters indicate the climatic variable affecting the species dynamics: T temperature, P precipitation, A aridity.

Supplementary Table 2). Given these results, it was possible that species overall sensitivity to climatic anomalies (either local or global) or lack of sensitivity (e.g., species for which density dependence was a main factor), also had a phylogenetic signal. Hence, we performed a post hoc analysis on the 143 species and found that phylogeny explained 88% of the species presence or lack of sensitivity to climatic anomalies (Supplementary Fig. 89 and Supplementary Table 2). The strong phylogenetic signal of both the degree of local adaptation (relative sensitivity to local-versus-global climate anomalies) and the overall total sensitivity to climatic anomalies, provides robustness to discard potential stochasticity in the results.

**The role of species phylogeny mediating local adaptation.** Our results suggest a dominant effect of evolutionary constraints (i.e., unmeasured traits with a strong phylogenetic signal) on the capacity for local adaptation. However, they provide no support for our expectation of low gene flow and high reproductive rates leading to increased adaptive potential.

An apparent degree of local adaptation shared by closely related species might be expected. Greater sensitivity to local climate anomalies implies spread and survival at sites where adaptation to local conditions is selected; this could be facilitated by the filtering of individuals from pre-existing genetic diversity or through phenotypic plasticity (e.g., behavioral, morphological or physiological attributes) favouring survival and eventual evolutionary adaptation. Closely related species share similar genetic and phenotypic attributes, although we did not find mediation of local adaptation through either mobility or voltinism. This is possibly because European butterfly species have resided in their current location for long periods (many butterflies expanded across Europe during their post-glacial colonization[37,38]), even creating refugia in Southern areas[39], resulting in a long-term established genetic differentiation currently poorly affected by mobility or reproductive rates, and hence with ample time to adapt to local conditions regardless of

reproductive rates. Note, that there was no effect of $G_{ST}$ when testing it as a proxy of genetic differentiation as an alternative to mobility (Supplementary Table 3). Phylogenetic signals have also been seen in distributional shifts of plant communities[40] and some butterfly species[41]; while mixed results have been found in relation to mobility and reproductive rate in range shifts or distributional trends[42,43]. Specific phylogenetic patterns of local adaptation might also result from the pressures of the external conditions affecting the capacity to evolve and adapt. For example, local adaptation might be more likely in certain regions or habitat types, with phylogenetic patterning in the associations of species to these conditions[44,45].

Globally adapted species could result from low local selective pressures, although we found no association between the overall variance in population dynamics explained by climate with species degree of local-versus-global adaptation ($P$ value = 0.54; Supplementary Fig. 90). Alternatively, global adaptation might result from a lack of genetic variation across the populations[46], or could be linked to high gene flow that is not reflected, or remained undetected, in our measure of species dispersal ability[47]. Extensive gene flow between populations and weak selection selective pressures may constrain local adaptation[48], though recent studies indicate that this is not always the case for gene flow[49,50]. Finally, global adaptation may result from pleiotropy, i.e., single genes affecting more than one phenotypic trait, limiting the degree of adaptive change due to selective trade-offs across the attributes for some species[51,52].

**Implications for ecological forecasting and conservation.** It is important to contextualize the intra- and interspecific variation of population fluctuations of butterfly species when facing climatic anomalies because they have implications for ecological fore-casting and conservation under climate change[9,53]. Local adaptations are likely to enhance stability for some populations under perturbations. For example, assuming similar niche breadth, locally adapted species at their retracting warm distributional

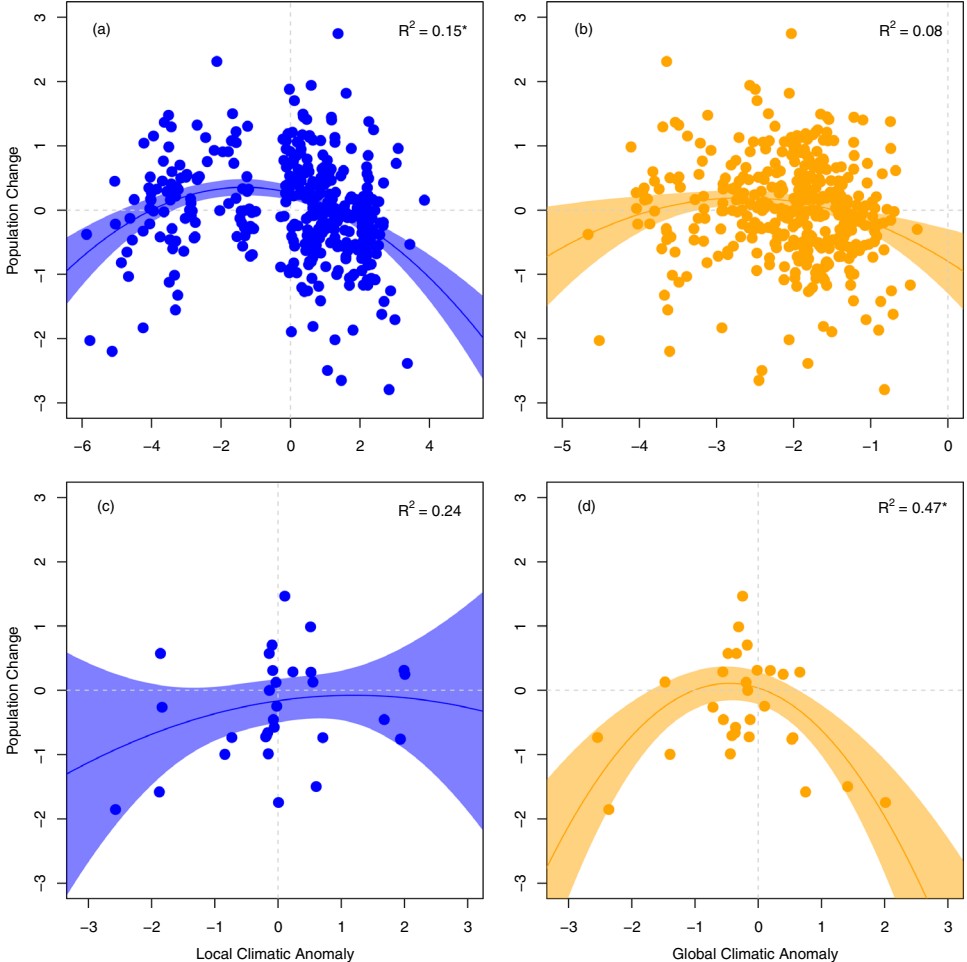

**Fig. 3 Population change in relation to local and global climatic anomalies. a, b** show local and global responses, respectively, for *Brenthis ino*, a species best adapted to local climatic anomalies in temperature during the overwintering period of the previous year of their adult stage ($t − 1$). **c, d** show local and global adaptation for *Cupido osiris*, a species whose population dynamics are best explained by global climatic anomalies in aridity during the pre-flight period of the year of their adult stage ($t$), indicating a lack of local adaptation. Colors indicate spatial scale (blue, local; orange, global), circles indicate raw data, lines show predicted trend with 95% confident intervals, asterisks indicate the best model for each species.

edges may remain within their fundamental niche space for longer than expected, compared with if they were globally adapted. This difference in species capacity to cope with environmental change could be of particular importance to the conservation of globally adapted species if populations at warm marginal edges contain the majority of the species genetic diversity[54]. In contrast, locally adapted populations at the colder leading edges of their ranges, expected to be strongholds of persistence under climate warming, may be more vulnerable to local heat extremes than if they were globally adapted. We suggest future research could seek to quantify if or how local adaptation interacts with the specific location of populations within their geographic range to influence population changes, also taking account of how climatic sensitivity within species can vary across their geographical range[55,56].

We acknowledge some limitations of the study, with some species (such as *Vanessa cardui*, Supplementary Fig. 86) showing unexpected strong signals of local adaptation despite multi-generational trans-continental migration of this species[57]. This might be due to spurious results but could be genuine and due to factors that we cannot elucidate within this study but are worth researching in the future; for example, the potential existence of several races with different environmental tolerances and dispersal patterns across Europe, or rapid adaptation to local

conditions over just one or two generations, potentially mediated by epigenetic changes[58]. Nonetheless, the relatively strong phylogenetic signal found here is contingent upon a few species with large leverage. Removing species based on data availability, or based on visualization of outliers in the tree, leads to a lower phylogenetic signal (Supplementary Fig. 91). We also do not discard the role of other factors such as habitat availability and connectivity in influencing butterfly species dynamics as well as potential interactions with the effect of the climatic anomalies, but their role was out of the scope of this study.

Despite raising interesting questions to pursue further, our overall results hold important implications for forecasting species responses to climate change. Namely, that local adaptation is important to factor into predictive climate models for a substantial subset of species, and this is likely to also influence advice regarding appropriate conservation practices in different locations.

## Methods
**Data collection**. Butterfly count data were extracted from three long-term Butterfly Monitoring Schemes carried out in Finland, Spain, and the UK, covering six out of the ten bioclimatic regions across Europe and three main biomes (Supplementary Figs. 92 and 93). The schemes consist of a network of sites where volunteers perform visual counts of butterflies along transects (i.e., sites) following the standardized "Pollard Walk" methodology[28]. Counts are conducted weekly during the

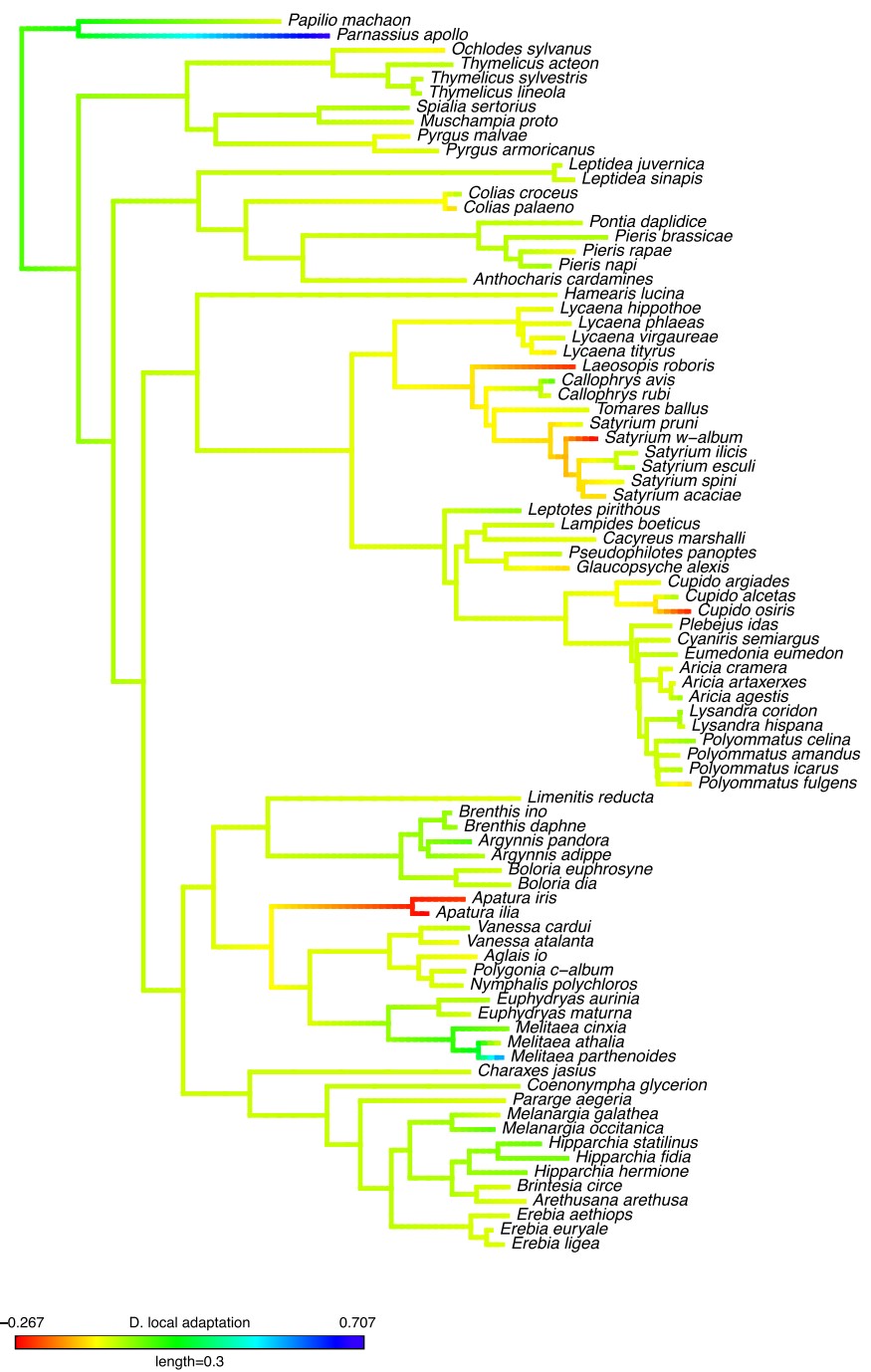

**Fig. 4 Phylogenetic comparative tree for the degree of local adaptation for the 86 species significantly affected by climatic anomalies.** Degree of local adaptation was set as a continuous distribution from −1 to 1. Negative (red) values of the degree of local adaptation relate to species adapted to climatic anomalies at the global scale, positive values (blue) the local scale.

butterfly flight season, which varies depending on the climatic zone within the range of March to the end of September. The species nomenclature is defined following Wiemers et al.[59].

For each site, year and species, we calculated an annual abundance index of adult butterflies that accounted for missing counts and spatial and temporal variation in species phenology[29,60]. We derived the annual species phenology from the observed weekly counts, fitting generalized additive models (GAM) for each bioclimatic region[29]. We then used the regional flight curve derived from the GAM as an offset in a generalized linear model to predict and impute values for missing counts. The complete series of weekly counts, including the imputed values, was then used to calculate an annual abundance index per species, site, and year. This method corrects for missing week counts, using the information derived from the regional phenology and produce indices that are less biased than indices derived from linear interpolations[29]. However, to assure consistent estimates, we excluded indices of abundance >50% of missing observations. We

used this annual species index of abundance ($N_i$) to calculate the interannual change in abundance as a proxy of the annual population growth rate. The rate was calculated per species population (i.e., per site) between two consecutive years as $\log(N_{t+1}/N_t)$.

The duration of the schemes (i.e., starting year) and the number of surveyed sites varied between countries: Finland (1999, $n_{sites} = 107$), Spain (1994, $n_{sites} = 130$) and the UK. (1976, $n_{sites} = 2128$). Thus, we used Finland (the limiting country) to set the range of study years (1999–2017) and the number of sites per country. To have sufficient interannual change data per site, we retained only sites having at least ten years of interannual abundance change data between 1999 and 2017. This rule reduced our sample to 53 sites for Finland, 59 for Spain and 701 for the UK. From the 701 UK sites, we used a random subset of 60 sites to balance the sample size between countries (Supplementary Figs. 94–99). These sites were selected randomly, apart from a condition maintaining at least 20 km separation between the selected sites.

As the effect of climate on population growth rates varies depending on the life stages of the organisms[61,62], we defined four life stages periods relevant for butterfly species[55]: pre-flight period, flight period, post-flight period, and overwintering. Pre-, post-, and flight periods vary across species, years, and bioclimatic zone[29]. Thus, we set these periods per species, year and bioclimatic zone[62], except the post-flight period of year $t$, which we discarded because it could have no possible effect on the population dynamics of the species in that year (adults in the count of that year had already perished). These periods were defined using the annual flight curve distribution from the relative abundances per species and zone (i.e., using the GAM fitted annual species phenology). The flight period was set as the dates between the 10th and 90th percentiles of the species distribution for each year and each bioclimatic zone. The pre-flight period was then set from February to the 10th percentile of the species flight period, and the post-flight period from the 90th percentile to the end of October. For multivoltine species (i.e., species with more than one reproduction per year), flight period was set between the 10th and 90th percentiles of the entire species flight-period distribution, independently of the generation. The overwintering period was set as fixed from November to January (of year $t – 1$ to year $t$), equally for all species and all sites. Although overwintering could be defined over different time periods, we fixed it to minimize potential overlaps between it and its adjacent time periods for some species in some regions[55].

We calculated climatic anomalies for temperature, precipitation, and aridity (as a combination of temperature and precipitation) because previous studies have demonstrated that butterflies, as also many other organisms, can be affected by each of these variables[21,33,34,63]. Climatic data were obtained from the European Climatic and Assessment Dataset project (ECAD)[64,65]. The data consisted of daily temperature (in degrees Celsius) and precipitation (in millimeters) from a gridded climatic dataset constructed from extrapolations of observations from a series of meteorological stations throughout Europe. We extracted the daily temperature and precipitation at 1° scale for each site for the period 1999–2017; and calculated local and global climatic anomalies for temperature and precipitation per species and phenological period (overwintering, pre-, post-, and flight period) as follows:

$$Wsi, local = Wsi − Ws \tag{1}$$

$$Wsi, global = Wsi − W \tag{2}$$

where $Wsi$ is the mean temperature or the total rainfall, depending on the climatic variable being analyzed per site ($s$) and year ($i$), $Ws$ is the mean temperature or the total rainfall per site for all studied years, and $W$ is the mean temperature or total rainfall for all studied sites and all years. Potential interactions between temperature and rainfall were calculated as the standardized aridity index (Eqs. (3) and (4))[33,66]:

$$SAI_{si,local} = −((P_{si} − P_s)/sdP_s) * 0.5(T_{si} − T_s)/sdT_s) \tag{3}$$

$$SAI_{si,global} = −((P_{si} − P)/sdP)) * 0.5(T_{si} − T)/sdT) \tag{4}$$

where $SAI$ stands for Standardized Aridity Index, $T$ for mean temperature, $P$ for total precipitation, $sd$ for standard deviation, with subindexes as above.

**Modeling population changes in relation to climatic anomalies.** We fitted three models to test whether species population growth rates were locally or globally adapted: a null model accounting only for density dependence to remove species not affected by the climatic anomalies when testing our hypotheses (Eq. (5)); a local model accounting for density dependence, the local anomaly term of the climatic variable being analyzed, and its quadratic term (Eq. (6)); and a global model with density dependence, the global anomaly of the climatic variable, and its quadratic term (Eq. (7)). Models were fitted independently per each species, with population growth rate as the response variable expressed as $log(N_{it}/N_{it−1})$. Site was set as a random intercept for all species, except for those with a low number of sites ($n = 23$ species), due to lack of model convergence when adding random effect; in these cases, only fixed effects were added (Supplementary Data 2). Errors were assumed to be normally distributed, and model diagnostics were used to test the conformation of the data to the model assumptions.

$$log(N_{it}/N_{it−1}) = N_{it−1} + ε \tag{5}$$

$$log(N_{it}/N_{it−1}) = N_{it−1} + W_{si,localt} + W^2_{si,localt} + ε \tag{6}$$

$$log(N_{it}/N_{it−1}) = N_{it−1} + W_{si,globalt} + W^2_{si,globalt} + ε \tag{7}$$

Where $N_i$ is the annual abundance index at the site $i$th and time $t$ or $t – 1$, $W_{it}$ is the climatic anomaly variable site $i$ at time $t$ either at the local or the global scale. Models including climate were fitted per species with all combinations of climatic variable $W$ (temperature, precipitation, and aridity), local or the global scale, time $t$ or $t − 1$ ($W_{it}$ and $W_{it-1}$, the latter not shown in the equations for simplicity), and all phenological periods (winter, pre-, post-, or flight period; except for the post-flight period at time $t$; Eqs. (6) and (7)):

This procedure resulted in a set of 43 models per species: the reduced model and 42 models consisting of each of the three climatic variables, per two spatial scales (local and global), per two time periods ($t$ and $t − 1$), and per four time periods (excepting the post-flight period at time $t$). From within these 43 models, we retained the best fitting model for each species for identifying the climatic

variable, spatial scale (local or global), and time ($t$ or $t − 1$) and phenological period, that most affected the species populations dynamics. Model selection was based on AIC value (Supplementary Data 2). Species with the null model as the best model were discarded from all further analyses (i.e., AIC null model—AIC alternative model <2; $n = 56$). We did this on the basis that in these cases our climatic variables explained almost nothing of the species population dynamics, i.e., the variance explained by models with climatic variables was close to null. We also discarded one species (*Polyommatus ripartii*) as the ΔAIC between models containing different climatic anomalies, different periods, times and scale differed <2; such as it was not possible to select the best model. We retained any species with the best models, including both the global and local anomalies of a specific climatic variable, at a specific period and time ($n = 11$). We did so because we were interested in the degree of local adaptation (see below); therefore both scales were eventually used. Model selection led to 86 species for which the best models included climatic anomalies at the local and the global scale, all with AIC lower than that of the null model. Following a conservative approach, we initially discarded nine out of these 86 species because they had less than ten years of interannual abundance change data (Supplementary Data 1 and Data 2). Discarding them did not influence the results related to attributes (see next section), but implied that some species were alone in their clade (e.g., *Apatura ilia*) or completely separated from the other groups (*Papilio machaon*). Therefore, we decided to add them back to have a more complete and less biased phylogenetic tree that encompassed all 86 species.

**The role of phylogeny and species attributes mediating the effect of climatic anomalies in species population dynamics.** For each species, we extracted the conditional $R^2$, i.e., the variance explained by the fixed effects without accounting for the variance due to random effects of the best fitting model and that of the model with the same climate variable but measured at the alternative spatial scale (local or global), keeping all other specifications (i.e., life-history period and year) fixed. We then used these variances to calculate a proxy of the degree of local adaptation per species by subtracting $R^2_{local\ model} − R^2_{global\ model}$. This proxy was subsequently used as the response variable to evaluate how local adaptation relates to mobility, voltinism, and phylogeny. To do so, we used a phylogenetic generalized linear model[67], with the species degree of local adaptation fitted to a normal error distribution, and mobility and voltinism as explanatory variables while controlling for the phylogenetic signal of the degree of local adaptation calculated using Pagel's lambda statistic[68].

Mobility and voltinism attribute values were retrieved from the literature[35,36,69], with voltinism categorized as either univoltine or multivoltine species (i.e., strictly one generation per year, and two or more generations per year respectively). Species that have a rare, or occasional, second generation were set as univoltine. The butterfly phylogenetic tree was extracted from the most recently published phylogenetic tree of European butterflies[37]. Alternative categorizations of voltinism were also tested with species categorized as univoltine, putative multivoltine or strict multivoltine, and categorized as per their know number maximum generations (1, 1.5, 2, or more than 2); we also included potential variability of voltinism set as the maximum and the minimum number of generation shown by a given species across Europe. Duration of flight period (in average months) and genetic differentiation ($G_{ST}$)[37] were also tested as alternative proxies of genetic differentiation and genetic flow (mobility). None of these alternatives differed qualitatively from the results of our initial model (Supplementary Table 3).

A post hoc analysis for the phylogenetic signal of the species sensitivity to climatic anomalies was done using D statistic[70]; i.e., whether climate was important in explaining the species population dynamics (best model included climatic anomalies, $N = 86$) or whether density dependence best explained the population dynamics (the null model was the best model).

We conducted all analyses in R 3.6.1 (R Core Team, 2019), using the package lme4[71] to fit our models and the package MuMIn[72] to estimate the marginal and conditional variances. Phylogenetic analyses were done using the phylosig and pgls functions in R package phytools[73].

## Data availability
The count and abundance butterfly data that support the findings of this study are available from the European Butterfly Monitor Scheme via a signed license agreement (https://butterfly-monitoring.net/). Climatic data are available via ECAD website (https://www.ecad.eu/).

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

## Acknowledgements

This work and Y.M. were supported by Marie Skłodowska Curie H2020-MSCA-IF-2017 795890 project EXTINCT of the European Commission; as well as, initially by Foundation for Biodiversity Research and CESAB (Centre for the Synthesis and Analysis of Biodiversity, France) project LOLA. We thank the eBMS (https://butterfly-monitoring.net/) for the data and their volunteers for providing the data needed for the study; and Roger Vila and the three reviewers for their revision on the manuscript.

## Author contributions

L.E., M.K., D.R., R.S., C.S., T.O., and Y.M. contextualized the study and its hypotheses. Y.M. led the paper and performed the analyses, with L.E. and supervised by T.O. L.E., M.K., D.R., R.S., C.S., T.O., and Y.M. all contributed to writing the manuscript.

## Competing interests

The authors declare no competing interests.
