## [Peer Review File · Communications Biology]

Reviewers' comments:

Reviewer #1 (Remarks to the Author):

Comments and reviewer recommendation for manuscript number COMMSBIO-21-0182-T "Local adaptation to climate anomalies relates to species phylogeny" by Melero et al.

This manuscript deals with an interesting topic, namely the possible local adaptation to climate anomalies in butterflies. The manuscript is well written and the tables and figures are very helpful. The analysis is appropriate and references are up-to-date. I only suggest some additional references (in the attached pdf), but the authors don't have to feel obliged to include them and a single comment on the possibility of underestimating species' mobility. An additional point of discussion could be that in the Finnish and UK monitoring sites, climate is probably more similar than in the Spanish monitoring scheme, where I presume there are sites that go from sea level to high up in the Pyrenees and thus have a much wider range of climates than Finland and/or the UK? To give an idea of the climate range, it would be good to give the range of the climate variables and climate anomalies in the different regions in a table or in the text somewhere.

Reviewer #2 (Remarks to the Author):

I have read with pleasure this study as it provides a clear evaluation of the responses of species to local versus wide climatic fluctuations and provides a simple model to search for possible correlations with species phylogeny and their traits. The records belong to the BMS effort representing one of the most precious datasets at the global scale. So, no doubts that this topic is of wide interest, based on strong data and thus it deserves publication on *Communications Biology*. However, despite the response variable (difference in R^2) has been assessed in a great details (I searched for any possible flaw in the methods), I feel that the predictors (traits) appear in comparison rather poor. The authors should enlarge their trait dataset and provide a more detailed comparative analysis. For example:

Voltinism:

Line 335-338 I think that the authors should make a distinction between strict univoltine species, strict multivoltine species and species being univoltine in some regions (north) and multivoltine in others (south). How a species univoltine in UK and multivoltine in Spain has been categorized? Moreover, there are species taking two years to develop (many *Erebia*). Are such kinds of species in the model? Another good variable could have been maximum-minimum number of generations, accounting for the ability of a given species to respond to local climate by changing the number of generations. This variable could be also included in the phylogenetic regression together with voltinism (I imagine the maximum over Europe has been used to detect multivoltine species). In my personal experience I also found that number of generations are a less strong predictor of genetic differentiation than length of flight period. Indeed, active dispersal of butterflies (thus gene-flow) is likely more determined by the number of days when a species is active instead of its generation number. While voltinism and flight period are strongly correlated, in some cases (e.g. *Vanessa atalanta* and *Gonepteryx rhamni*) univoltine species can fly for 8-10 months per year, with a high potential for gen-flow. Detailed data for voltinism (maximum and minimum), flight period, together with many other traits have been recently published for European butterflies and one of the authors of this study is also author of the database <https://www.nature.com/articles/s41597-020-00697-7>. Here, other potential interesting traits can be found (trophic generalism, wing size, behavioural traits) and they could help in filling the gap highlighted by the author themselves at line 118 "Our results suggest a dominant effect of evolutionary constraints (i.e. unmeasured traits with a strong phylogenetic signal) on the capacity for local adaptation".

Gene flow and mobility:

Line 129 and 143. "This is possibly because mobility is too coarse a measure of potential gene flow" "A more direct evaluation of gene-flow among populations for all these species is currently available in a study already cited from the authors (Dapporto et al 2019 *Mol Ecol*) as a measure of G_{st} among populations in their COI. Still not a definitive measure of gene-flow but maybe a better

proxy than mobility

The dataset is freely available at:

<https://datadryad.org/stash/dataset/doi:10.5061/dryad.2q76p8f>
in the file "tabfinal.txt"

Range size:

why not to include range size in Europe which is available for all species and it could be highly correlated with climatic plasticity. Although trivial, such a relationship can be used to show that more widely distributed species are (for example) more globally adapted than locally. Range size is available in the climber variables recently updated and available on dryad here
<https://datadryad.org/stash/dataset/doi:10.5061/dryad.hmgqnk9dh>

Another point I don't understand is why the authors did not examine local and global R2 independently and then their difference. This could reveal if local and global responses (not their relative values) relate with specific traits.

Moreover, I also have some doubts about the definition of "adaptation". Adaptation is usually referred as the process by which a species becomes fitted to its environment in general as a result of natural selection acting upon heritable variation. I think that in this study the authors have mostly assessed phenotypic plasticity (the possibility to develop different phenotypes depending on the environment faced by an individual/populations). Indeed, the authors have modelled "adaptation" as the capability to adjust phenology traits according to differing temperatures. So, in this case "adaptation" is the capability to show different phenotypes according to the local conditions. This is clear to me by the definition at line 99-101.

Still in this vein

Line 306-309. Why to discard species showing no dependence with climate? They could represent a nice group (not plastic at any scale) to be compared (as a first step) against the others for example by using a phylogenetic ANOVA (the geiger R package has a nice function for this)

The fact that local "adaptation" has a phylogenetic signal is very interesting to me and provides robustness to the index calculated by the authors. Indeed, there could be a strong noise in their calculation, but such a strong phylogenetic signal provides a good evidence to discard stochasticity (they could discuss this to make their assessment stronger).

Line 160. I'm sorry to cite another paper of mine (for the third time) but the existence of southern richness and northern purity in European butterflies has been just published in Communications Biology and this could help in discussing this section of the discussion.

<https://www.nature.com/articles/s42003-021-01834-7>

As a final comment. How the species list has been assessed? I imagine that the recent Wiemers et al (2019) check list has been the main taxonomic reference but (for example) I can see that *M. athalia* and *M. celadussa* appear to be merged. I could accept lumping very close (quasi-species?) taxa in order to obtain a better assessment of their phenotypic plasticity (*P. malvae-malvodes*, *P. edusa-daplidice* etc) but the authors should clarify how their taxonomic list has been obtained.

With my congratulations for this nice study
Leonardo Dapporto

Reviewer #3 (Remarks to the Author):

Review of ms titled « Local adaptation to climate anomalies relates to species phylogeny" by Melero and colleagues for Communications Biology.

General comments

This study aims at correlating demographic changes in European butterfly species over the last 20 years with variation in local and global climatic trends and "anomalies" for temperature and precipitation. It is indeed needed to model animal demographic trends beyond mere global climatic

trends that represent changes in averages (and ignore the increasing rate of anomalies such as extreme temperature events) and are obtained as averages for large geographical areas (and ignore the important effect of microclimatic variation on population responses). Yet, I have several major concerns that should be tackled by the authors to justify the relevance of the work and the robustness of their conclusions. Some of these concerns may be due to the fact that I, myself, do not model such types of data and some explanations may be enough to provide a relevant rebuttal to my concerns.

One concern is about how the authors quantify "local adaptation". Local adaptation is quantified by the authors as a "greater sensitivity" (hence stronger R^2 for the model that links demographic trends and local climatic variation) of demographic trends to local weather anomalies (as described lines 66-67, and lines 80-81). But this is counterintuitive: local adaptation of butterfly populations would lead to maintenance of high butterfly abundance **DESPITE** climatic variation and anomalies. Although I find it interesting to quantify the effect of local and global climatic variation and anomalies on demographic trends, I do not agree that a higher correlation with local climatic conditions would be taken as evidence (as a "proxy") for local adaptation.

A second major problem I have with the models that are used is that they do not tease apart (mal)adaptation to climate anomalies versus to other environmental factors that rapidly changing due to human activities, such as decreased habitat availability and connectivity, or increasing pesticide use in agricultural fields, to name two other factors affecting butterfly demography to a much larger extent than climate warming. This is not trivial because climate warming is not the main factor responsible for the demographic crash of butterfly species (and other insects), and because the other, more important, environmental factors show correlated changes with climate. Hence, the models that the authors use and that link climate and demography are problematic by attributing demographic crashes to climate change while the causal link is likely another, correlated, variable such as decreased habitat suitability through increased urbanization, increase in pesticide use in fields.... This is not a specific problem of this paper, as the whole field of research on climate warming has the same approach, but since the publication of the IPBES 2019 global report, it has become urgent that this field of research acknowledge the general weakness of the models used and try and move forward using models that tease apart the effect of climate and of other, much more important, changes in natural environments, to explain the current biological hemorrhage we are facing. Otherwise, we will have the wrong conservation targets and populations will continue to crash.

Third, I have an issue (in italics and bold) regarding the second prediction of the authors: "Therefore, we expected greater sensitivity to local weather anomalies for i) *less mobile species*, and ii) *species with higher reproductive rates* (assessed in terms of the number of generations per year)". Why not lower? More individuals, due to higher reproductive rates, means more genetic diversity, which means more variation on which selection can act: adaptation is indeed accelerated when effective size increases, not the opposite.

Fourth, we should have access for the test of hypothesis 1 to the actual R^2 values for the models including local and global climatic variation, for each species, which is what is used to test the authors' first hypothesis (sorry if I miss these R^2 values somewhere in the manuscript). These R^2 values are likely low, in addition to the fact (discussed above) that other environmental factors correlate with the average trends in climate change (warming and precipitation) but are not found in the models.

Regarding the test of the second hypothesis that assesses the role of phylogenetic history on adaptation to climate variation, I have two major comments.

Fifth, the figure 3 tests the second hypothesis about the phylogenetic history explaining the level of local versus global adaptation in the butterfly species. However, I wonder whether the analyses hold without a couple of outlier species, as can be seen in Figure 3: such as *Parnassius io* or the *Patura* genus (2 species)? I doubt it. Could you please show the analyses without these three species to quantify the relative contribution of these three (outlier?) species on the general impression that phylogenetic history explains the level of climatic sensitivity of butterflies in general.

Sixth, I do not understand either why the value " $R^2_{\text{local model}} - R^2_{\text{global model}}$ " was used as a proxy and response variable to quantify local adaptation, in the phylogenetic analyses. R^2 values inform about the correlation between climatic variables and demographic trends, so the higher the

R², the higher the correlation. What is the rationale of subtracting R² of global climatic variation from that of the local climatic variation as a variable informing about adaptation? This is actually using as a proxy of local adaptation a different quantity than was had been used for testing hypothesis 1. And it makes not more sense to me (see my earlier comment about how to quantify "local adaptation" for your hypothesis 1, above).

More minor comments:

The authors write that different "sensitive phenological periods" were tested. Reading the MM, one can see that in addition to the "flight period", wintering is one of the phenological periods taken into account in this paper, but it is unclear how. It would be worth explaining what has been done here, and also rephrase the conclusions about the flight period, as I doubt that flight period is the most sensitive phenological period for facing climatic variation by butterflies. Rather wintering is likely the main stage at which natural selection reduces populations sizes to very low numbers, but usually we have much less data regarding wintering than flight period. Please rephrase to avoid misleading conclusions about the importance of the flight period on the demographic sustainability of butterfly species.

Line 94 : you mention that the null model "density-dependance" explained best the data for a large part of the dataset (57 species out of 147, that is more than 1/3 of the species). Could you thus develop: it deserves an explanation as why this was chosen as the null model and what biological explanation this models suggests tod demographic changes in butterfly populations.

Regarding the second hypothesis of phylogenetic history on local/global adaptation, The MM should be including more detailed explanations of how the analyses were conducted in practise.

Response to Reviewers' comments – COMMSBIO-21-0182-T

Dear reviewers,

Please check answers to the comments below.

Yolanda Melero on behalf of all the authors.

Reviewer #1 (Remarks to the Author):

Comments and reviewer recommendation for manuscript number COMMSBIO-21-0182-T "Local adaptation to climate anomalies relates to species phylogeny" by Melero et al.

This manuscript deals with an interesting topic, namely the possible local adaptation to climate anomalies in butterflies. The manuscript is well written and the tables and figures are very helpful. The analysis is appropriate and references are up-to-date. I only suggest some additional references (in the attached pdf), but the authors don't have to feel obliged to include them and a single comment on the possibility of underestimating species' mobility.

We are grateful for the positive comments of the reviewer which are very encouraging. We have added the mentioned references and the minor corrections annotated in the pdf. For example, we have removed the "Cop" in the reference number 31 (previously 30), since it was an error from the referencing software. Now it reads as:

Stefanescu, C., Carnicer, J. & Peñuelas, J. Determinants of species richness in generalist and specialist Mediterranean butterflies: the negative synergistic forces of climate and habitat change. *Ecography*. 34, 353–363 (2011).

In relation to annotated comment "another possibility is that mobility is underestimated .." please be aware that we have now followed advice from reviewer 2 and also tested our hypotheses with other proxies of gene flow (i.e. G_{st} , average number of flight months; see Supplementary Table 5), with no different results; which supports the finding that traits related to gene flow, as with mobility, have limited predictive capacity.

An additional point of discussion could be that in the Finnish and UK monitoring sites, climate is probably more similar than in the Spanish monitoring scheme, where I presume there are sites that go from sea level to high up in the Pyrenees and thus have a much wider range of climates than Finland and/or the UK? To give an idea of the climate range, it would be good to give the range of the climate variables and climate anomalies in the different regions in a table or in the text somewhere.

We have now added a boxplot summarizing annual mean climatic variables per country in a Supplementary Figure (Supplementary Figure 8), where it is possible to see the variation among the studied countries for the three climatic variables (temperature, precipitation and aridity). We agree this information helps to visualize variation between countries and climatic variables. The figure and its legend reads as follows:

Supplementary Figure 8. Boxplot for the annual mean (a) temperature, (b) precipitation, and (c) aridity for the monitoring sites located in Spain, Finland and United Kingdom for the period 1999-2017.

Reviewer #2 (Remarks to the Author):

I have read with pleasure this study as it provides a clear evaluation of the responses of species to local versus wide climatic fluctuations and provides a simple model to search for possible correlations with species phylogeny and their traits. The records belong to the BMS effort representing one of the most precious datasets at the global scale. So, no doubts that this topic is of wide interest, based on strong data and thus it deserves publication on Communications Biology. However, despite the response variable (difference in R^2) has been assessed in a great details (I searched for any possible flow in the methods), I feel that the predictors (traits) appear in comparison rather poor. The authors should enlarge their trait dataset and provide a more detailed comparative analysis. For example:

Voltinism: Line 335-338 I think that the authors should make a distinction between strict univoltine species, strict multivoltine species and species being univoltine in some regions (north) and multivoltine in others (south). How a species univoltine in UK and multivoltine in Spain has been categorized?

We thank the reviewer for these positive comments and advice. We have followed most of the detailed suggestions here and below, and commented those that we have not. In relation to the predictors, we have performed new analyses with the recommended variables that relate to voltinism and mobility (see specific answers below). However, we decided not to include new different types of traits to our analysis. We did so because we have specific *a priori* hypotheses related to traits affecting adaptation via gene flow and reproductive rates. Therefore, analysis of traits such as trophic generalism and range size, although potentially interesting, we feel would broaden the paper unnecessarily and detract from the focus on our specific hypotheses, and may also lead to spurious associations from multiple tests being carried out.

Specifically, with regards to voltinism we have now added some of the suggested tests. However, to explain our initial approach: we chose to classify species were as either strict univoltine (those with a single generation throughout their range) or putative multivoltine (species with two or more generations whenever conditions are favorable), rather than a splitting multivoltine species into strict multivoltine species and those that are sometimes univoltine. This is because multivoltinism depends strongly on the environmental conditions, especially weather, and so populations of putative multivoltine species showing univoltinism might be circumstantial and variable depending on the annual climate. Hence, species may be partially or strict multivoltine contingent on the (varying) environmental conditions during the sampling period and sampling extent rather than a strict definition based on capability from species' life history constraints. Species with 1.5 generations (i.e. partial bivoltine) were also

categorized as univoltine since the extra generation are mostly occasional and local. This is, for instance, the case in N Spain, where several species behave as univoltine in areas in the Pyrenees only because there is a thermic constraint to complete a second generation. However, the same populations could behave as bivoltine under unusual favorable circumstances.

Notwithstanding this, to satisfy the reviewers query we have now also tested voltinism as a categorical variable with the advised levels: strict univoltine, putative multivoltine and strict multivoltine, but no qualitative differences in our results were found. To double check, we also tested voltinism categorized as a categorical variable per number of generations: 1, 1.5, 2 or >2, and again results were not significant. These results have now been added in Supplementary Table 5.

Moreover, there are species taking two years to develop (many *Erebia*). Are such kinds of species in the model? Whilst these species' life-cycles may take two years in sites where biennial life-cycles occur, adults appear annually; since our population changes are based on adult counts changes should be detected annually, not biennially. We account that for some locations adults of these species might be affected by the generation of two previous years. The importance of this time lag, is however, accounted for in our model related for the overwintering period in time $t-1$ (please see Supplementary Table 1 and Supplementary Table 2).

Another good variable could have been maximum-minimum number of generations, accounting for the ability of a given species to respond to local climate by changing the number of generations. This variable could be also included in the phylogenetic regression together with voltinism (I imagine the maximum over Europe has been used to detect multivoltine species).

We agree that the ability to shift voltinism can be an interesting variable to check, so we have now also analyzed voltinism and max-min of generations, with the first categorized as per our categories (univoltine vs multivoltine) and as also advised by the reviewer (univoltine, putative multivoltine and strict multivoltine) based on the provided published database, and the second added as an extra numerical variable. No significant differences were found, so we preferred to leave the main MS simple and keep these extra results for the supplementary material (see Supplementary Table 5).

In my personal experience I also found that number of generations are a less strong predictor of genetic differentiation than length of flight period. Indeed, active dispersal of butterflies (thus gene-flow) is likely more determined by the number of days when a species is active instead of its generation number. While voltinism and flight period are strongly correlated, in some cases (e.g. *Vanessa atalanta* and *Gonepteryx rhamni*) univoltine species can fly for 8-10 months per year, with a high potential for gen-flow. Detailed data for voltinism (maximum and minimum), flight period, together with many other traits have been recently published for European butterflies and one of the authors of this study is also author of the database <https://www.nature.com/articles/s41597-020-00697-7>.

Thanks for this suggestion. We have now also tested duration of flight period as a proxy of voltinism, as well as duration of flight period and max-min number of generations. No different results were found so this was also added to Supplementary Table 5. Note, there is also the issue that some long-living species (e.g. *G. rhamni*) can enter in diapause as an adult during their adult stage, hence their flight period duration.

Here, other potential interesting traits can be found (trophic generalism, wing size, behavioural traits) and they could help in filling the gap highlighted by the author themselves at line 118 "Our results suggest a dominant effect of evolutionary constraints (i.e. unmeasured traits with a strong phylogenetic signal) on the capacity for local adaptation".

Please see response above. We agree that might be interesting but none of these traits were part of our *a priori* hypotheses relating local adaptation with gene flow and reproductive rates (voltinism and mobility). Thus, we hope the reviewer appreciates our preference to avoid *post-hoc* hypotheses to the MS which would detract from the focus on our specific hypotheses, and may also lead to spurious associations from multiple tests being carried out.

Gene flow and mobility:

Line 129 and 143. "This is possibly because mobility is too coarse a measure of potential gene flow " A more direct evaluation of gene-flow among populations for all these species is currently available in a study already cited from the authors (Dapporto et al 2019 Mol Ecol) as a measure of G_{st} among populations in their COI. Still not a definitive measure of gene-flow but maybe a better proxy than mobility. The dataset is freely available at:

<https://datadryad.org/stash/dataset/doi:10.5061/dryad.2q76p8f>
in the file "tabfinal.txt"

We agree G_{st} can be a good proxy of genetic flow between populations. Thus, we have also now tested it and mention this briefly in the main text. Yet, as with mobility, results were not significant and effect sizes were similar. Hence, we decided to retain our *a priori* predictors but add these new results in Supplementary Table 5. We tested G_{st} with voltinism (as per our category) and max-min number of generations (see Supplementary Table 5).

In relation to these new analyses detailed above we added the following text to material and methods (lines 406-413): "Alternative categorizations of voltinism were also tested with species categorized as univoltine, putative multivoltine or strict multivoltine, and categorized as per their known maximum generations (1, 1.5, 2 or more than 2); we also included potential variability of voltinism set as the maximum and minimum number of generations shown by a given species across Europe. Duration of flight period (in average months) and genetic differentiation (Dapporto et al. 2019) were also tested as alternative proxies of genetic differentiation and genetic flow (mobility). None of these alternatives differed qualitatively from the results of our initial model (Supplementary Table 5)."

Range size: why not to include range size in Europe which is available for all species and it could be highly correlated with climatic plasticity. Although trivial, such a relationship can be used to show that more widely distributed species are (for example) more globally adapted than locally. Range size is available in the climber variables recently updated and available on Dryad

here <https://datadryad.org/stash/dataset/doi:10.5061/dryad.hmgqkngdh>

As previously commented, we felt that we should keep our hypotheses rather than adding new ones. We agree on the potential interest of this predictor, yet we think that rather than range size per se the position within the species distributional range could be a more important factor (i.e. leading, trailing or center) in relation to their response to the local climatic anomalies. Having shown the existence of a degree of local adaptation within this current study, position in the range is something that we are currently working on for our next paper.

Another point I don't understand is why the authors did not examine local and global R^2 independently and then their difference. This could reveal if local and global responses (not their relative values) relate with specific traits. Analyzing the association of local and global R^2 independently in relation to traits would essentially be asking the extent to which a species population dynamics can be explained by weather in relation to their traits. We would note that many such analyses have been carried out for European butterflies (e.g. Roy et al. 2008; WallisDeVries et al. 2011; McDermott Long et al. 2017; Mills et al. 2017). The key novelty in the current study is how the difference in variance explained by local and global climate anomalies is indicative of degree of local adaptation. Hence, we retain this key focus in the analysis.

Moreover, I also have some doubts about the definition of "adaptation". Adaptation is usually referred to as the process by which a species becomes fitted to its environment in general as a result of natural selection acting upon heritable variation. I think that in this study the authors have mostly assessed phenotypic plasticity (the possibility to develop different phenotypes depending on the environment faced by an individual/populations). Indeed, the authors have modelled "adaptation" as the capability to adjust phenology traits according to differing temperatures. So, in this case "adaptation" is the capability to show different phenotypes according to the local conditions. This is clear to me by the definition at line 99-101.

Phenotypic plasticity involves gene-by-environment interactions (for example, oviposition microsite preferences varying between locations depending on the local macroclimate). As the reviewer correctly outlines, this would allow a buffering of the effects on temperature variation on populations. The result of this is that species would have a broader climatic tolerance, i.e. the curves in Figure 1b or 1f (previously set as Supplementary Fig 1) would be broader and shallower, meaning there is a greater range of temperatures in which population growth can remain positive. Although this is adaptation of a sort, it is different to the accepted definition of evolved local adaptation whereby the optimum of a thermal performance differs amongst the populations of species so that they perform best in their 'home' conditions. We have now made this clearer in the figure legend. Given the importance of this figure for our hypotheses, we have moved it from the Supplementary to the main text as Figure 1. We hope this figure now clearly shows how our metric assesses degree of evolutionary adaptation to local conditions (i.e. as found in a better fit of population dynamics to local weather anomalies versus global anomalies). Please note, that phenotypic plasticity may be exhibited across the entire range, or it might only occur in certain areas, i.e. there might conceivably be local adaptation of the phenotypic plasticity. Alternatively, local adaptation can occur in fixed traits, such as lighter-coloured insects in warmer areas (Zeuss et al. 2014). We have also now noted this in Figure 1 legend.

Still in this vein. Line 306-309. Why to discard species showing no dependence with climate? They could represent a nice group (not plastic at any scale) to be compared (as a first step) against the others for example by using a phylogenetic ANOVA (the *geiger* R package has a nice function for this)

We agree on the interest of testing for differences between the two groups: weather sensitive and not weather sensitive. Using *phytools* package (to be consistent with our methodology) we have done a phylogenetic generalized linear regression to test the role of voltinism and mobility of whether a species is sensitive or not to the climate (fit a binary response variable). No effect was found for the traits (p -value > 0.08) but a strong phylogenetic signal was also found here with D -statistic = 0.88. We have now included a post-hoc analysis of the phylogenetic signal (see main text, lines 117-123) as well as the phylogenetic tree of the classification (Supplementary Fig. 4; consisting of the phylogenetic tree for the sensitivity of species to climatic anomalies as a

binary categorical variable, N = 143 species) with corresponding phylogenetic signal value (D = 0.88; Supplementary Table 4).

The following text was also added in Material and Methods (lines 414-418):

"A post-hoc analysis for the phylogenetic signal of the species sensitivity to climatic anomalies was done using D statistic (Münkemüller et al. 2012); i.e. whether climate was important in explaining the species population dynamics (best model included climatic anomalies, N = 86) or whether density dependence best explained the population dynamics (the null model was the best model)."

The fact that local "adaptation" has a phylogenetic signal is very interesting to me and provides robustness to the index calculated by the authors. Indeed, there could be a strong noise in their calculation, but such a strong phylogenetic signal provides a good evidence to discard stochasticity (they could discuss this to make their assessment stronger).

In relation to this, and in continuation to the previous comment, we have added (lines 121-123):

"The strong phylogenetic signal of both, the degree of local adaptation and the sensitivity to climatic anomalies, provides robustness to discard potential stochasticity in the results."

Line 160. I'm sorry to cite another paper of mine (for the third time) but the existence of southern richness and northern purity in European butterflies has been just published in Communications Biology and this could help in discussing this section of the discussion. <https://www.nature.com/articles/s42003-021-01834-7>

Thank you. We have modified the text which now reads (lines 135-141):

"This is possibly because European butterfly species have resided in their current location for long periods (many butterflies expanded across Europe during their post-glacial colonization (Hewitt 1999; Dapporto et al. 2019), even creating refugia in Southern areas (Dincă et al. 2021), resulting in a long-term established genetic differentiation currently poorly affected by mobility or reproductive rates, and hence with ample time to adapt to local conditions regardless of reproductive rates. Note, that there was no effect of GST when testing it as a proxy of genetic differentiation as an alternative to mobility (Supplementary Table 5)."

As a final comment. How the species list has been assessed? I imagine that the recent Wiemers et al (2019) check list has been the main taxonomic reference but (for example) I can see that *M. athalia* and *M. celadussa* appear to be merged. I could accept lumping very close (quasi-species?) taxa in order to obtain a better assessment of their phenotypic plasticity (*P. malvae-malvoides*, *P. edusa-daplidice* etc) but the authors should clarify how their taxonomic list has been obtained.

Yes, we used Wiemers et al 2019, but *M. celadussa* was not included in the selected sites. The species occurs only in Catalonia amongst the study areas, but the selected sites of the CBMS did not include *M. celadussa*. We have now added to the text (lines 268-269): "The species nomenclature is defined following Wiemers et al (2019)".

With my congratulations for this nice study

Leonardo Dapporto

Thanks for the helpful review!

Reviewer #3 (Remarks to the Author):

General comments

This study aims at correlating demographic changes in European butterfly species over the last 20 years with variation in local and global climatic trends and "anomalies" for temperature and precipitation. It is indeed needed to model animal demographic trends beyond mere global climatic trends that represent changes in averages (and ignore the increasing rate of anomalies such as extreme temperature events) and are obtained as averages for large geographical areas (and ignore the important effect of microclimatic variation on population responses). Yet, I have several major concerns that should be tackled by the authors to justify the relevance of the work and the robustness of their conclusions. Some of these concerns may be due to the fact that I, myself, do not model such types of data and some explanations may be enough to provide a relevant rebuttal to my concerns.

One concern is about how the authors quantify "local adaptation". Local adaptation is quantified by the authors as a "greater sensitivity" (hence stronger R^2 for the model that links demographic trends and local climatic variation) of demographic trends to local weather anomalies (as described lines 66-67, and lines 80-81). But this is counterintuitive: local adaptation of butterfly populations would lead to maintenance of high butterfly abundance DESPITE climatic variation and anomalies. Although I find it interesting to quantify the effect of local and global climatic variation and anomalies on demographic trends, I do not agree that a higher correlation with local climatic conditions would be taken as evidence (as a "proxy") for local adaptation.

We acknowledge there might be some difficulty in the interpretation of our hypotheses, and in the reasoning on local adapted species being more sensitive to local climatic anomalies, hence we have now attempted to clarify this. In the previous Supplementary Figure 1 (now moved to the main text as Fig. 1 for its importance for our hypotheses), we used a simulation to show how local adaptation to temperature (i.e. multiple thermal performance curves across different populations, with each population finding optimum growth closer to local mean conditions; Figure 1a) leads to greater variance in population dynamics explained by- (and greater sensitivity to-) local weather anomalies (e.g. please compare Figure 1b with Fig. 1c, and see the opposite pattern for a hypothetical species with a single global thermal performance curves across all populations in panels d, e and f). We have now further developed the legend of this figure to explain these patterns.

Also, please be aware that we do not test mean abundances *per se*, which indeed might be higher for local adapted species, rather we analysed *interannual change* in abundance.

The Figure 1 was included in the initial submission as Supplementary Material, but please find it also below including our legend now with substantial amendments:

Figure 1. Simulations of the consequences of global and local adaptation on the population responses to local and global climatic anomalies. Panels a and d show the performance of species at five sites with climatic means spanning across the range of the climatic variable. We expected locally adapted species to present multiple different performance curves representing distinct populations at sites distributed along the species' distributional range, as shown in panel a. This expectation implies that population change will be more sensitive to local weather anomalies (simulation in panel b) than to weather anomalies calculated from all sites across the species' distribution (simulation in panel c). In the case of global adaptation, performance is represented by a single curve through its entire range (panel d). Therefore, observed population change will be more sensitive to global weather anomalies calculated from all sites across the species' distribution (panel f) than to the local site anomalies (panel e). Performance curves were based on a Briere type I function (Briere et al. 1999), which is a simple function that matches empirical data on thermal performance (Shi and Ge 2010). We included a fixed area under the curve as consistent with expectations of specialist/generalist tradeoffs (Angilletta et al. 2003). Beyond this hypothetical example, in practice, the mean and variance of curves may vary across species; for example, some species may have broader climatic tolerances than others, i.e. the curves in panels b or f would be broader and shallower,

meaning there is a greater range of temperatures in which population growth can remain positive. Broader tolerances may be driven in part by phenotypic plasticity, i.e. gene-by-environment interactions (for example, oviposition microsite preferences varying between locations depending on the local macroclimate). This phenotypic plasticity may be exhibited across the entire range, or it might only occur in certain areas, i.e. there might conceivably be local adaptation of the phenotypic plasticity. Alternatively, local adaptation can occur in fixed traits, such as lighter-coloured insects in warmer areas (Zeuss, 2014). Both of these evolutionary adaptations produce patterns akin to that in panel a, whereby optimum of a thermal performance differs amongst the populations of species so that they perform best in their 'home' conditions. To generate weather across the range, we standardized an observed 19-year time-series of global yearly temperatures (min =0, max=1, mean=0.5) and then shifted the values of each year to predict mean expectation at local sites across the range, a local value for each site and year was then sampled with Gaussian noise. The performance was subsequently used as input into a discrete logistic growth model ($N_{t+1} = RN_t(1 - N_t/K)$) as proportional to R the intrinsic growth rate. Each population was seeded with a small number of individuals and was allowed to recover by immigration should the population size go to zero. A time series of population change for each of the sites was collected from the simulation (ΔN after initialization and immigration was excluded). Models for population change were then fitted using local and global anomalies and are shown in b, c, e and f. Colors in panels a and d indicate location in the distributional (e.g. blue to red, cold to hot extremes respectively). Colors in the rest of panels indicate spatial scale (blue- local climate anomalies; orange- global climate anomalies), circles indicate populations (i.e. from distinct sites), lines show predicted trend with 95% confident intervals.

A second major problem I have with the models that are used is that they do not tease apart (mal)adaptation to climate anomalies versus to other environmental factors that rapidly changing due to human activities, such as decreased habitat availability and connectivity, or increasing pesticide use in agricultural fields, to name two other factors affecting butterfly demography to a much larger extent than climate warming. This is not trivial because climate warming is not the main factor responsible for the demographic crash of butterfly species (and other insects), and because the other, more important, environmental factors show correlated changes with climate. Hence, the models that the authors use and that link climate and demography are problematic by attributing demographic crashes to climate change while the causal link is likely another, correlated, variable such as decreased habitat suitability through increased urbanization, increase in pesticide use in fields....al)adaptation to climate anomalies versus to other environmental factors that rapidly changing due to human activities, such as decreased habitat availability and connectivity, or increasing pesticide use in agricultural fields, to name two other factors affecting butterfly demography to a much larger extent than climate warming. This is not trivial because climate warming is not the main factor responsible for the demographic crash of butterfly species (and other insects), and because the other, more important, environmental factors show correlated changes with climate. Hence, the models that the authors use and that link climate and demography are problematic by attributing demographic crashes to climate change while the causal link is likely another, correlated, variable such as decreased habitat suitability through increased urbanization, increase in pesticide use in fields....

This is not a specific problem of this paper, as the whole field of research on climate warming has the same approach, but since the publication of the IPBES 2019 global report, it has become urgent that this field of research acknowledge the general weakness of the models used and try and move forward using models that tease apart the effect of climate and of other, much more important, changes in natural environments, to explain the current biological hemorrhage we are facing. Otherwise, we will have the wrong conservation targets and populations will continue to crash.

We are aware of the importance of other factors in the population trends of butterfly species, and biodiversity, in general. Yet, the fact that other perturbations are also affecting species demography does not diminish the importance of understanding the role of climate change, and most importantly, climatic anomalies on the species dynamics. Weather variation is known to have a fundamentally importance in driving population dynamics of

butterflies (e.g. Fernández-Chacón et al. 2014; Mills et al. 2017; Herrando et al. 2019), but so far no study has analyzed the role of local adaptation on species responses to these anomalies in real-world field conditions. There are of course also other intrinsic factors such as density dependence, which we control for in our models. However, our study focuses on interannual changes, which are unlikely driven by long gradual term changes like habitat fragmentation and chemical use associated with intensive farming, although we appreciate that local land use can mediate responses to climatic extremes and have developed work on this ourselves (e.g. Oliver and Morecroft 2014; Oliver et al. 2015).

We agree on the importance of investigating the role of other potential factors, and we actually are currently researching the effect of environmental heterogeneity in potential interaction with climate change, but first we need to address the degree of local adaptation to climatic change by the species. Understanding with some confidence how perturbations affect populations is not an easy task and while models are an imperfect interpretation of reality, they offer information needed to solve research uncertainties; yet this comes with limitations; for example, adding many factors (explanatory variables) in a model can easily overparametrize it, resulting in a non-understandable or weak model unless it is provided by a huge amount of data. Therefore, although we appreciate the reviewers point, we feel this does not invalidate our approach to quantify interspecific variation in local adaptation to climate. It is important to note, that other drivers such as habitat and land use are unlikely to have confounding effects in our analysis, since in each country the sampling sites cover a broad range of different land use types. Nonetheless, we have added an additional note on the limitations of our study in relation to extending analysis to interacting effects with climate (lines 185-188).

Third, I have an issue (in italics and bold) regarding the second prediction of the authors: "Therefore, we expected greater sensitivity to local weather anomalies for i) less mobile species, and ii) species with higher reproductive rates (assessed in terms of the number of generations per year)". Why not lower? More individuals, due to higher reproductive rates, means more genetic diversity, which means more variation on which selection can act: adaptation is indeed accelerated when effective size increases, not the opposite.

Our hypothesis is in the same direction as suggested by the reviewer here: we expect that higher reproductive rates lead to *higher* evolutionary adaptive capacity (due to more sexual selection increasing genetic novelty). If higher reproductive rates from more generations per year are also associated with higher population numbers (which is possible, but not always true, for example if there are density-dependent limits to population size), then this too would increase genetic variance upon which selection can act. We have now reworded the text to improve clarity and it now reads (lines 66-68): "Therefore, we expected increased adaptation to local weather anomalies for i) less mobile species, and ii) species with higher reproductive rates (assessed in terms of the number of generations per year)." Please also check Fig. 1 for a complementary description of our hypotheses.

Fourth, we should have access for the test of hypothesis 1 to the actual R^2 values for the models including local and global climatic variation, for each species, which is what is used to test the authors' first hypothesis (sorry if I miss these R^2 values somewhere in the manuscript). These R^2 values are likely low, in addition to the fact (discussed above) that other environmental factors correlate with the average trends in climate change (warming and precipitation) but are not found in the models.

We agree on the importance of accessibility of results, and all R^2 values for all species and all model combinations are in the Supplementary Table 1, where each row represents the species x model specification (null model; or models for time: *t* and *t-1*, period: *OW*, *pre-FP*, *FP* and *post-FP*, and climatic variable: temperature, precipitation and aridity) at the local and global scales. Columns relate to AIC values, and R^2 of each model combination (R^2 marginal and total in the case of random effect models). The variance explained in by the R^2 global and local were not expected to be high since, as commented by the reviewer, there are other non-climatic factors driving population change, as well as other aspects of weather not perfectly captured by our models. Please note our response to the comment above, however, that we are working on modelling *interannual* population changes, so environmental factors correlating with long term trends in climate are unlikely to be confounding factors in our analysis here since increases in warming are rarely linear and have high variability about mean trends. Please note also that we excluded species from further analysis for which we could not find a significant relationship with weather variables. We have now also conducted an additional analysis to test for phylogenetic signal in this capacity for weather variables to explain population dynamics (lines 117-123 and lines 414-418; Supplementary Fig. 4 and Supplementary Table 4).

Regarding the test of the second hypothesis that assesses the role of phylogenetic history on adaptation to climate variation, I have two major comments.

Fifth, the figure 3 tests the second hypothesis about the phylogenetic history explaining the level of local versus global adaptation in the butterfly species. However, I wonder whether the analyses hold without a couple of outlier species, as can be seen in Figure 3: such as *Parnassius io* or the *Patura* genus (2 species)? I doubt it. Could you please show the analyses without these three species to quantify the relative contribution of these three (outlier?) species on the general impression that phylogenetic history explains the level of climatic sensitivity of butterflies in general.

Thanks for this insight. We have done the analyses as requested and, reviewer is correct in pointing out that phylogenetic signal will drop, i.e. Pagel's Lambda decreases from 0.84 to 0.085. this is included in a new Supplementary Fig. 6 (also shown below). However, note that while *Parnassius apollo* might be an outlier in relation to its degree of local adaptation (R_2), *Apatura* genus' degree of adaptation is similar to that of other several other species (for example see *Satyrrium w-album*, *Cupido Osiris* and *Laeosopis roboris* in Figure 2 of the manuscript). Hence, *Apatura* is not really an outlier in terms of local adaptation. If the reviewer refers as them being outlier in relation to the phylogenetic tree, *Apatura* species are neither outliers here to us, since other species/genuses also stand alone such as e.g. *Charaxes jasius*. We understand that visually in the tree *Apatura* species are both together and have similar values of local adaptation, but it is exactly this what one would expect when there is a phylogenetic relationship in local adaptation.

Overall, we do not see a clear objective criteria for removing the requested species. A criteria for removing species in the phylogenetic tree is not an easy task, we also tested removing those species with less of 10 data points (see lines 380-384), leaving 77 species remaining and a Pagel's lambda of 0.0007, but then some other species became phylogenetic outliers left aside in the tree (e.g. *P. machaon*, *Apatura ilia*, *Callirhytis rubi*). We think the approach is, rather than removing species, to follow a less conservative yet more complete approach, including all species affected by climatic variation. However, to address the reviewers concerns we have now added supplementary analysis showing how removing species based on both data availability (e.g. those with less than 10 data points), and based on visualization of outliers in the tree (e.g. removing the three species suggested by the reviewer), leads to lower phylogenetic signal (Supplementary Fig. 6) and we mention this contingency in the results in the main text (lines 183-185)

Supplementary Figure 6. Phylogenetic comparative trait plots of the butterfly traits for the species degree local adaptation, mobility (both with a continuous distribution) and voltinism (with a discrete distribution), for (a) the 77 species with Nsample size > 10, Pagel's lamda = 0.0007; and b) the 83 species left after removing visual outliers: *Parnassius apollo*, *Apatura ilia* and *Aptaura iris*, Pagel's lamda = 0.085. Negative (red) values of the degree of local adaptation relate to species adapted to climatic anomalies at the global scale, positive values (blue) the local scale.

Sixth, I do not understand either why the value "R2local model – R2global model" was used as a proxy and response variable to quantify local adaptation, in the phylogenetic analyses. R2 values inform about the correlation between climatic variables and demographic trends, so the higher the R2, the higher the correlation. What is the rationale of subtracting R2 of global climatic variation from that of the local climatic variation as a variable informing about adaptation? This is actually using as a proxy of local

adaptation a different quantity than was had been used for testing hypothesis 1. And it makes not more sense to me (see my earlier comment about how to quantify “local adaptation” for your hypothesis 1, above).

Hopefully this issue is clarified by our revised Figure 1 legend (please see it shown above in response to comments from Reviewer #2). We subtracted R^2 local - R^2 global because our focus was the comparative importance of the scale of the species adaptation, i.e. the aim of our research was test for the existence of a continuum degree of adaptation to weather from species with populations more adapted to the climate anomalies occurring as assessed at the particular site (local climatic anomaly) to others responding to anomalies calculated across the whole species distribution (global climatic anomalies). Most species will likely respond to both but in a different degree/sensitivity and, hence we wanted to check how much species are locally adapted (versus globally) and how this adaptation is mediated by traits. As mentioned, please see adapted Figure 1 legend for the logic behind this. This is something that, except for a few experimental (and mainly single-species) studies, has not been previously tested in the wild; contrary to the effect of weather on species to micro-to-macro scaled climatic events (e.g. (Mills et al. 2017; Trisos et al. 2020) and the role of different species traits (e.g. Haeeler et al. 2014; Pacifici et al. 2017).

Both hypotheses H₁ (existence of a continuum degree of local adaption) and H₂ (the mediation of mobility and voltinism) were tested using the proxy for the species degree of local adaptation (R^2 local – R^2 global). This is mentioned in the main text (lines 98-99): “The degree of apparent local adaptation for each species was calculated as the difference between the two R^2 values, ranging from -1 to 1, with higher values indicating more local adaptation” and in the MM (lines 390-396).

More minor comments:

The authors write that different “sensitive phenological periods” were tested. Reading the MM, one can see that in addition to the “flight period”, wintering is one of the phenological periods taken into account in this paper, but it is unclear how. It would be worth explaining what has been done here, and also rephrase the conclusions about the flight period, as I doubt that flight period is the most sensitive phenological period for facing climatic variation by butterflies. Rather wintering is likely the main stage at which natural selection reduces populations sizes to very low numbers, but usually we have much less data regarding wintering than flight period. Please rephrase to avoid misleading conclusions about the importance of the flight period on the demographic sustainability of butterfly species.

In the manuscript we specify that four phenological periods were taken in account: overwintering, pre-flight period, flight period and post-flight period. This was done following previous evidence that the key period for the effect on climate change on abundances and population dynamics varies between species. For some species it may be the overwintering period (which may impact on larva/pupa survival for some species), for others the flight period (which may impact on reproduction and mortality). For other species the larval stage (pre-flight period) is most important. This has already been proven (e.g. Roy et al. 2008; WallisDeVries et al. 2011; Radchuk et al. 2013; Mills et al. 2017), and therefore it is not part of our objectives and hypotheses neither our conclusions, it was rather a necessary step in our analyses to test for the effects of weather on multiple possible life history periods (as the reviewer suggests and as also done elsewhere, e.g. Mills et al; 2017).

We had not intended to inadvertently imply that weather during the flight period was all important. To make it clearer how the periods were defined and why, we have now added more references to the text that explain and support this, and slightly modified the text to make our calculation of the overwintering period clearer (Lines 308-311), where now it reads: “The overwintering period was set as fixed from November to January (of year t-1 to year t), equally for all species and all sites. Although overwintering could be defined over different time periods, we fixed it to minimize potential overlaps between it and its adjacent time periods for some species in some regions”.

Line 94 : you mention that the null model “density-dependance” explained best the data for a large part of the dataset (57 species out of 147, that is more than 1/3 of the species). Could you thus develop: it deserves an explanation as why this was chosen as the null model and what biological explanation this models suggests tod demographic changes in butterfly populations.

Density dependence (DD) is an important fundamental factor affecting the growth rate of many species’ populations. It has been researched intensely for decades and there are many available fundamental papers and books showing that the abundance of the previous year(s) affect that of the next and the overall growth and stability of the populations (e.g., Vandermeer and Goldberg 2004). Note, that the apparent strength of density dependence can be inflated by sampling errors (cf. regression to the mean), but this in no way obviates the need to control for it’s effect as a major source of variation in measured population dynamics. Hence, we set the null model including DD as an explanatory variable to control for it. We removed those species for which this null model was the best fit using model selection based on AIC (i.e. for these species there was no evidence that other models including climate variables were a better fit). For 43% of species the climatic anomalies we assessed did not have a detectable effect on their dynamics. On reflection, we now feel it was inappropriate to state that the density dependence was the main factor explaining the population dynamics of these species, since this was a null model and we didn’t fit a control model without DD (because our focus was to control for it not understand it’s relative importance). Hence, we have removed this statement. To address the reviewers request about justifying the

importance of controlling for density dependence we have now inserted a referenced statement into the main text (lines 75-77).

Regarding the second hypothesis of phylogenetic history on local/global adaptation, The MM should be including more detailed explanations of how the analyses were conducted in practise.

We have now changed the text that now reads (see full paragraph in lines 396-400): "To do so, we used a phylogenetic generalized linear model, with the species degree of local adaptation fitted to a normal error distribution, and mobility and voltinism as explanatory variables while controlling for the phylogenetic signal of the degree of local adaptation calculated using Pagel's lambda statistic.", and later (lines 419-422) "Phylogenetic analyses were done using functions `phylosig` and `pgls` in R package `phytools`".

References

- Angilletta MJ, Wilson RS, Navas CA, James RS (2003) Tradeoffs and the evolution of thermal reaction norms. *Trends Ecol. Evol.* 18:234–240
- Briere J-F, Pracros P, Le Roux A-Y, Pierre J-S (1999) A Novel Rate Model of Temperature-Dependent Development for Arthropods. *Environ Entomol* 28:22–29. <https://doi.org/10.1093/ee/28.1.22>
- Dapporto L, Cini A, Vodă R, et al (2019) Integrating three comprehensive data sets shows that mitochondrial DNA variation is linked to species traits and paleogeographic events in European butterflies. *Mol Ecol Resour* 19:1623–1636. <https://doi.org/10.1111/1755-0998.13059>
- Dincă V, Dapporto L, Somervuo P, et al (2021) High resolution DNA barcode library for European butterflies reveals continental patterns of mitochondrial genetic diversity. *Commun Biol* 4:1–11. <https://doi.org/10.1038/s42003-021-01834-7>
- Fernández-Chacón A, Stefanescu C, Genovart M, et al (2014) Determinants of extinction-colonization dynamics in Mediterranean butterflies: the role of landscape, climate and local habitat features. *J Anim Ecol* 83:276–285. <https://doi.org/10.1111/1365-2656.12118>
- Haeler E, Fiedler K, Grill A (2014) What Prolongs a Butterfly's Life?: Trade-Offs between Dormancy, Fecundity and Body Size. *PLoS One* 9:e111955. <https://doi.org/10.1371/journal.pone.0111955>
- Herrando S, Titeux N, Brotons L, et al (2019) Contrasting impacts of precipitation on Mediterranean birds and butterflies. *Sci Rep* 9:1–7. <https://doi.org/10.1038/s41598-019-42171-4>
- Hewitt GM (1999) Post-glacial re-colonization of European biota. In: *Biological Journal of the Linnean Society*. Academic Press, pp 87–112
- McDermott Long O, Warren R, Price J, et al (2017) Sensitivity of UK butterflies to local climatic extremes: which life stages are most at risk? *J Anim Ecol* 86:108–116. <https://doi.org/10.1111/1365-2656.12594>
- Mills SC, Oliver TH, Bradbury RB, et al (2017) European butterfly populations vary in sensitivity to weather across their geographical ranges. *Glob Ecol Biogeogr* 26:1374–1385. <https://doi.org/10.1111/geb.12659>
- Münkemüller T, Lavergne S, Bzeznik B, et al (2012) How to measure and test phylogenetic signal. *Methods Ecol Evol* 3:743–756. <https://doi.org/10.1111/j.2041-210X.2012.00196.x>
- Oliver TH, Marshall HH, Morecroft MD, et al (2015) Interacting effects of climate change and habitat fragmentation on drought-sensitive butterflies. *Nat Clim Chang* 5:941–946. <https://doi.org/10.1038/nclimate2746>
- Oliver TH, Morecroft MD (2014) Interactions between climate change and land use change on biodiversity: attribution problems, risks, and opportunities. *Wiley Interdiscip Rev Clim Chang* 5:317–335. <https://doi.org/10.1002/wcc.271>
- Pacifici M, Visconti P, Butchart SHM, et al (2017) Species' traits influenced their response to recent climate change. *Nat Clim Chang* 7:205–208. <https://doi.org/10.1038/nclimate3223>
- Radchuk V, Turlure C, Schtickzelle N (2013) Each life stage matters: the importance of assessing the response to climate change over the complete life cycle in butterflies. *J Anim Ecol* 82:275–285. <https://doi.org/10.1111/j.1365-2656.2012.02029.x>
- Roy DB, Rothery P, Moss D, et al (2008) Butterfly numbers and weather: predicting historical trends in abundance and the future effects of climate change. *J Anim Ecol* 70:201–217. <https://doi.org/10.1111/j.1365-2656.2001.00480.x>
- Shi P, Ge F (2010) A comparison of different thermal performance functions describing temperature-dependent development rates. *J Therm Biol* 35:225–231. <https://doi.org/10.1016/j.jtherbio.2010.05.005>
- Trisos CH, Merow C, Pigot AL (2020) The projected timing of abrupt ecological disruption from climate change. *Nature* 580:496–501. <https://doi.org/10.1038/s41586-020-2189-9>
- Vandermeer JH, Goldberg DE (2004) Population ecology: first principles. *Choice Rev Online* 41:41-3429-41-3429. <https://doi.org/10.5860/choice.41-3429>
- WallisDeVries MF, Baxter W, van Vliet AJH (2011) Beyond climate envelopes: Effects of weather on regional population trends in butterflies. *Oecologia* 167:559–571. <https://doi.org/10.1007/s00442-011-2007-z>
- Zeuss D, Brandl R, Brändle M, et al (2014) Global warming favours light-coloured insects in Europe. *Nat Commun* 5:1–9. <https://doi.org/10.1038/ncomms4874>

REVIEWERS' COMMENTS:

Reviewer #1 (Remarks to the Author):

The revised version of this manuscript has included most if not all comments of the reviewers. I have no further comments on this version and can only congratulate the authors with this nice work!

Reviewer #2 (Remarks to the Author):

It is a pleasure to read this revised version of the manuscript. The authors have done a great job in improving their analysis according to my comments (and to the comments of the other reviewers). I still have a doubt however about the term "adaptation". Indeed, I imagine that a species which is best adapted to a local climate and to its fluctuations does not vary too much its frequency among years, and in this respect it should show no correlation and no "adaptation" according to the index used by the authors. Conversely, species which are not very well adapted to the full local climate range could show strong fluctuations in case of unfavorable years and thus a high "adaptation" index. In my opinion the index they used is more a sort of climatic "sensitivity", which in some respect, is a very interesting index in the optic of current and future climate changes. I'm I wrong? If the authors think that this doubt is plausible (species with high adaptation index are species "suffering" for climatic fluctuations, they could provide some elucidation in the first part of their study.

Then, I read the MS and I only have a very minor comment

Line 55 usually studies using species as cases (that I like a lot) are indicated as "comparative". I suggest: "In this study, we performed a comparative analysis using multiple species of butterflies as a study system because,.."